# Activity enhancement of cobalt catalysts by tuning metal-support interactions

Carlos Hernández Mejía [ID] [1], Tom W. van Deelen [ID] [1] & Krijn P. de Jong[1]

Interactions between metal nanoparticles and support materials can strongly influence the performance of catalysts. In particular, reducible oxidic supports can form suboxides that can decorate metal nanoparticles and enhance catalytic performance or block active sites. Therefore, tuning this metal-support interaction is essential for catalyst design. Here, we investigate reduction-oxidation-reduction (ROR) treatments as a method to affect metal-support interactions and related catalytic performance. Controlled oxidation of pre-reduced cobalt on reducible ($TiO_2$ and $Nb_2O_5$) and irreducible ($\alpha$-$Al_2O_3$) supports leads to the formation of hollow cobalt oxide particles. The second reduction results in a twofold increase in cobalt surface area only on reducible oxides and proportionally enhances the cobalt-based catalytic activity during Fischer-Tropsch synthesis at industrially relevant conditions. Such activities are usually only obtained by noble metal promotion of cobalt catalysts. ROR proves an effective approach to tune the interaction between metallic nanoparticles and reducible oxidic supports, leading to improved catalytic performance.

---

[1] Inorganic Chemistry and Catalysis, Debye Institute for Nanomaterials Science, Utrecht University, Universiteitsweg 99, 3584 CG Utrecht, The Netherlands. These authors contributed equally: Carlos Hernández Mejía, Tom W. van Deelen. Correspondence and requests for materials should be addressed to K.P.d.J (email: k.p.dejong@uu.nl)

Metal particles catalyze major chemical processes, including the synthesis of bulk chemicals and transportation fuels[1–4]. The catalytic reactions that are involved occur mainly at the metal surface. Therefore, the surface-to-volume ratio of these particles is typically maximized through nanosizing, leading to adequate exposure of the active sites and efficient utilization of the metal[5,6]. Supports play a critical role in stabilizing the reactive nanoparticles but also in generating new interphase phenomena, such as electron transfer or shape rearrangement[7–9]. These phenomena can have ambivalent consequences for the catalytic performance such as promoting activity and selectivity or blocking active sites, which decreases the catalyst efficiency.

A renowned example of support effects occurs when reducible metal oxide supports ($TiO_2$, $Nb_2O_5$, $CeO_2$, etc.) are partially reduced by hydrogen spillover from the metal nanoparticle[10], i.e., the so-called strong metal–support interaction (SMSI)[11,12]. As a result, suboxide species are generated from the support and these may migrate onto the metal nanoparticles surface[13–16]. In the case of cobalt catalysts for the Fischer–Tropsch (FT) synthesis, i.e., the conversion of synthesis gas ($H_2$/CO) into ultra clean fuels and chemicals, the effect of SMSI is twofold. On one hand, the cobalt catalysts benefit from the mildly Lewis acidic character of the suboxide species, as it increases the intrinsic activity (turnover frequency, TOF) and selectivity towards $C_{5+}$ products[17,18]. The increased TOF is believed to originate from the interface between the metal and suboxide where the CO-dissociation rate is enhanced[19–21]. On the other hand, the catalytic activity can be lowered by a dense overlayer of suboxide species that block the access to the active sites[14]. The interplay between promotion of the catalytic performance and blockage of active sites is not clear and control over the metal-support interactions may thus provide an opportunity to increase the number of accessible, promoted active sites.

The SMSI state can be reversed by oxidative treatments. However, this reversibility has been mainly studied for noble metals, while reports of base metals are scarce[12,22–24]. Soled et al.[25] studied the impact of SMSI on $H_2$-chemisorption for cobalt nanoparticles supported on titania. The $H_2$-uptake was initially suppressed because of site blockage but increased significantly after a reduction–oxidation–reduction (ROR) treatment. This brought $H_2$-chemisorption and TEM and XRD results into closer agreement, but the effect of ROR on the structure of the cobalt particles as well as its implications for catalysis remained unclear. Besides reversal of the SMSI state, ROR has been applied to regenerate deactivated catalysts through redispersion of the active metal[26–29]. The work on catalyst regeneration focused, however, on cobalt catalysts supported on irreducible oxides, such as $Al_2O_3$ and $SiO_2$, in which SMSI effects do not play a role.

In this work, we investigate the possibility to tune the metal-support interactions of cobalt supported on reducible metal oxides via reduction-oxidation-reduction treatments with the aim to enhance the catalytic activity in the Fischer–Tropsch synthesis. To this end, $Co/TiO_2$ and $Co/Nb_2O_5$ are synthesized and subjected to ROR treatments. Co on irreducible $\alpha$-$Al_2O_3$ is prepared for comparison. After the first reduction, oxidation above 100 °C leads to the formation of hollow cobalt oxide particles. Upon re-reduction, the $H_2$-uptake doubles for the reducible metal oxides due to optimization of the SMSI state and not by cobalt redispersion, while the $H_2$-uptake decreases for $\alpha$-$Al_2O_3$. The modified metal-support interaction on reducible supports drastically enhances the catalytic activity under industrially relevant FT conditions, effectively doubling the number of available active sites without changing their TOF. Modulation of the metal–support interaction via ROR is thus a promising strategy to boost the catalytic performance and to greatly improve the efficiency of metal catalysts supported on reducible oxides.

## Results

**Reduction.** Cobalt particles supported on reducible oxides $TiO_2$ and $Nb_2O_5$, and on irreducible oxide $\alpha$-$Al_2O_3$ were prepared by incipient wetness impregnation followed by drying and thermal treatment. The nominal cobalt loading was 6 wt% and experimentally determined loadings are given in Supplementary Table 1. $H_2$-chemisorption analysis was performed on the pristine samples, which were reduced at 350 °C prior to the measurement (referred to as R-samples). After $H_2$-chemisorption, the samples were passivated by exposure to air at room temperature and subsequently analyzed using scanning transmission electron microscopy-energy-dispersive X-ray spectroscopy (STEM-EDX, Fig. 1a–c). The cobalt particles were rather uniformly distributed over $TiO_2$ and $Nb_2O_5$ while the spatial distribution was more heterogeneous on $\alpha$-$Al_2O_3$.

$H_2$-chemisorption and STEM-EDX results for the R-samples are given in Table 1. The theoretical $H_2$-uptake was calculated for cobalt particles using the surface-average particle diameter derived from STEM-EDX images. This theoretical value was compared to the experimental uptake measured by $H_2$-chemisorption. In the case of R-$TiO_2$ and R-$Nb_2O_5$, a substantial discrepancy in $H_2$-uptake was observed between both methodologies. This is explained by partial coverage of the metallic cobalt surface by suboxide species leading to suppressed uptake in the $H_2$-chemisorption measurement. It is important to note that the total amount of suboxide species is probably low, because no overlayers could be detected with STEM-EDX nor with in situ TEM of $Co/Nb_2O_5$ under reducing conditions in a previous study[30]. No large discrepancy between the theoretical and experimental $H_2$-uptake of R-$Al_2O_3$ was observed, because suboxides were not formed.

**Reduction–oxidation.** After the first reduction treatment at 350 °C, the R-samples were subjected to an oxidative treatment at temperatures varying between 30 and 400 °C (referred to as RO30- to RO400-samples). The temperature-programmed reduction (TPR) profiles of pristine $Co_3O_4/Nb_2O_5$ and RO30-$Nb_2O_5$ to RO400-$Nb_2O_5$ are shown in Fig. 2a. The pristine sample displayed two-step, consecutive reduction from $Co_3O_4$ to CoO (280–300 °C), and from CoO to Co (300–400 °C). Samples that had been reduced and oxidized at temperatures up to 100 °C showed only one sharp reduction peak at 230 °C corresponding to the reduction of CoO. Therefore, the second reduction of these samples can be performed at substantially lower temperature than the first one. Based on the amount of consumed $H_2$, we found that some metallic cobalt was still present on those samples, probably as a metallic core covered by a CoO layer. Further increasing the oxidation temperature to 150 °C led to broadening of the narrow peak (RO150-$Nb_2O_5$) and eventually resulted in a two-step reduction profile, associated with the formation of $Co_3O_4$ during oxidation (RO200-$Nb_2O_5$). Upon oxidation at 400 °C, the reduction profile resembled that of the pristine $Co_3O_4/Nb_2O_5$ sample.

The RO-samples were characterized by STEM-EDX mapping. Samples oxidized at low temperatures appeared unchanged (Fig. 2b, c). However, at oxidation temperatures of 200 °C, hollow cobalt oxide particles were observed (Fig. 1d–f). The formation of these hollow particles was ascribed to the Kirkendall effect[31–33]. During oxidation, an initial oxide layer forms around the metallic particle and at higher temperature, the smaller $Co^{2+}$ or $Co^{3+}$ cations diffuse outwards through the oxide layer faster than the larger $O^{2-}$ anions diffuse inwards, forming shells of

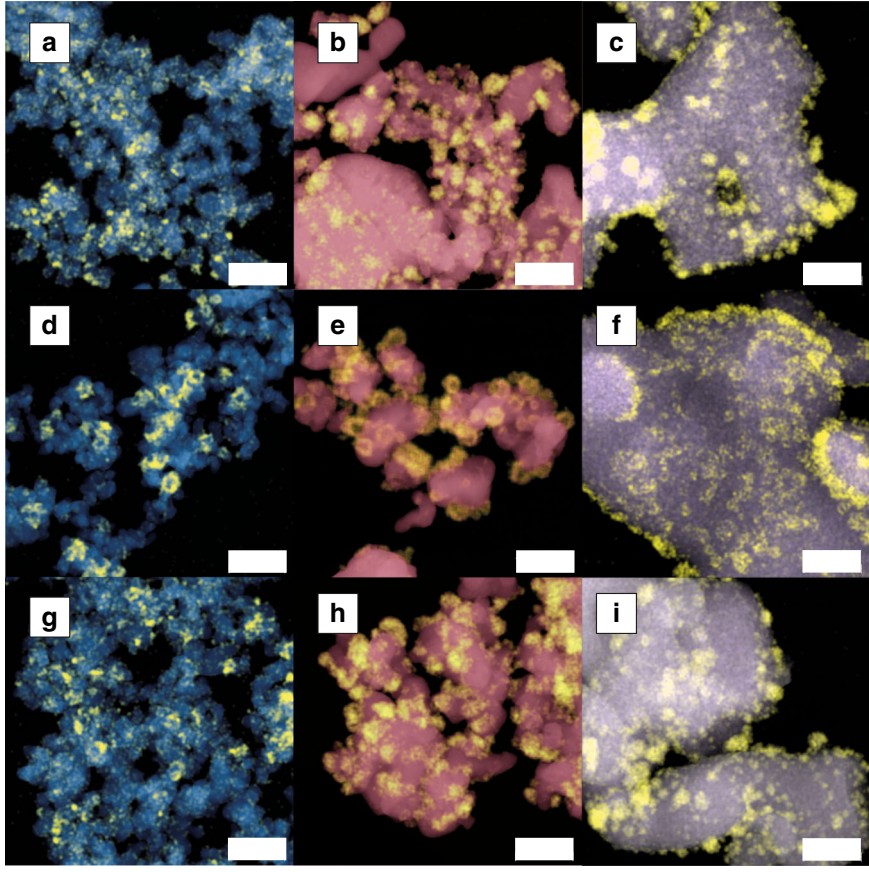

**Fig. 1** Electron microscopy images of the samples upon reduction-oxidation-reduction treatments. STEM-EDX mapping of $Co/TiO_2$, $Co/Nb_2O_5$, and $Co/\alpha$-$Al_2O_3$ at various stages of the ROR treatment. Co is depicted in yellow, Ti in blue, Nb in red, and Al in violet. **a–c** Samples after reduction at 350 °C and passivation (R-samples). **d–f** Samples after reduction at 350 °C and oxidation at 200 °C (RO200-samples). **g–i** Samples after reduction at 350 °C, oxidation at 200 °C, and reduction at 220 °C (RO200R-samples) and passivation. The scale bars corresponds to 100 nm. Additional electron microscopy results for these samples can be found in Supplementary Figures 1–3

---

**Table 1 The results of cobalt particle size analyses using STEM-EDX and $H_2$-chemisorption for the pristine samples after reduction at 350 °C (R-samples) and the samples after reduction at 350 °C, oxidation at 200 °C, and reduction at 220 °C (RO200R-samples)**

| Sample | Particle diameter $D[3,2]\pm$standard deviation (nm)[a] | Theoretical $H_2$-uptake ($\mu mol_{H_2} \cdot g_{Co}^{-1}$)[a] | Experimental $H_2$-uptake ($\mu mol_{H_2} \cdot g_{Co}^{-1}$)[b] |
|---|---|---|---|
| *R-samples* | | | |
| R-TiO$_2$ | 11 ± 3.4 | 787 | 341 |
| R-Nb$_2$O$_5$ | 19 ± 10 | 437 | 265 |
| R-Al$_2$O$_3$ | 13 ± 6.5 | 666 | 631 |
| *RO200R-samples* | | | |
| RO200R-TiO$_2$ | 10 ± 3.8 | 865 | 783 |
| RO200R-Nb$_2$O$_5$ | 18 ± 7.9 | 470 | 657 |
| RO200R-Al$_2$O$_3$ | 21 ± 11 | 412 | 499 |

[a]Derived from STEM-EDX
[b]Derived from $H_2$-chemisorption

---

cobalt oxide leaving behind empty cores. At an oxidation temperature of 400 °C, the hollow cobalt oxide particles got distorted and agglomerated.

**Reduction–oxidation–reduction.** The samples after an ROR cycle were characterized by $H_2$-chemisorption (Fig. 3a). The second reduction of the ROR treatment was performed at 220 °C, whereas the R-samples had been reduced at 350 °C. For the

reducible oxides, the $H_2$-uptake significantly increased at oxidation temperatures above 100 °C. In the case of $TiO_2$, it changed from 341 $\mu mol_{H_2} g_{Co}^{-1}$ for R-$TiO_2$ to 783 $\mu mol_{H_2} g_{Co}^{-1}$ for RO200R-$TiO_2$. Similarly, for $Nb_2O_5$ it increased from 265 $\mu mol_{H_2} g_{Co}^{-1}$ for R-$Nb_2O_5$ to 657 $\mu mol_{H_2} g_{Co}^{-1}$ for RO200R-$Nb_2O_5$. On $\alpha$-$Al_2O_3$ however, a substantial decrease of $H_2$-uptake was observed after ROR treatments. The degree of reduction (DOR) was determined by TPR directly after the last reduction for all R-

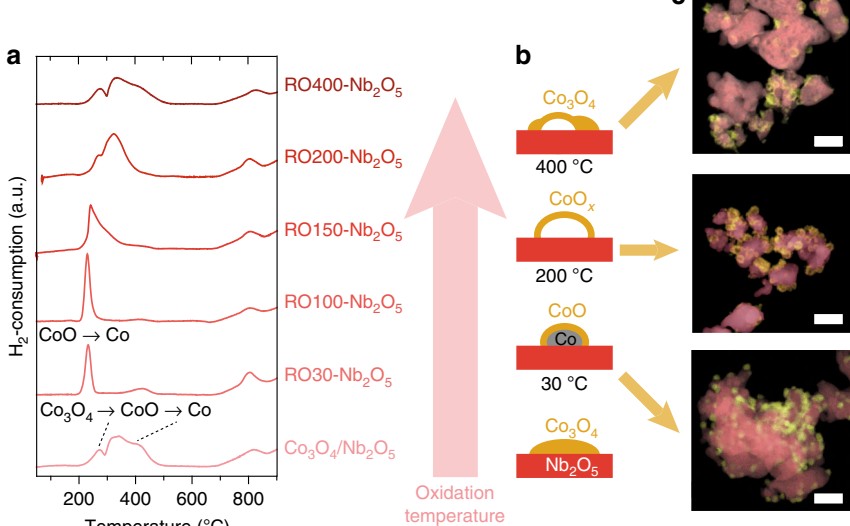

**Fig. 2** The effect of the oxidation temperature on the structure of Co/Nb$_2$O$_5$. **a** Temperature-programmed reduction profiles, **b** schematic illustration, and **c** STEM-EDX images of Co/Nb$_2$O$_5$ after reduction and oxidation at various temperatures (30–400 °C). Co is depicted in yellow, Nb in red. The scale bar corresponds to 100 nm

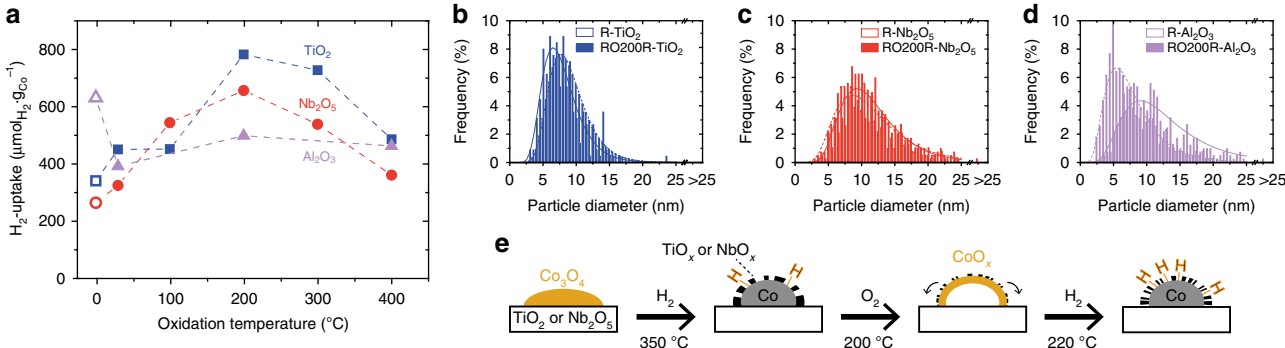

**Fig. 3** The effect of ROR treatments on the exposed cobalt surface area and particle size. **a** Hydrogen uptake for Co/TiO$_2$ (blue squares), Co/Nb$_2$O$_5$ (red circles), and Co/α-Al$_2$O$_3$ (violet triangles) after various reduction–oxidation–reduction treatments, as determined by H$_2$-chemisorption. The results of the pristine samples after reduction at 350 °C (open symbols) are included at 0 on the x-axis. **b–d** Particle size distributions of cobalt supported on TiO$_2$, Nb$_2$O$_5$, and α-Al$_2$O$_3$, respectively, after reduction at 350 °C (open bars) and after reduction at 350 °C followed by oxidation at 200 °C and reduction at 220 °C (solid filled bars), as determined by STEM-EDX. **e** Schematic illustration of the effect of the ROR treatment on cobalt supported on reducible oxides

and RO200R-samples and for RO400R-Nb$_2$O$_5$ (Supplementary Figure 4 and Supplementary Table 2). The DOR for the R-samples was 97–98% and decreased slightly to 84–95% after ROR. Because these changes were minor, the H$_2$-chemisorption results were not corrected for the DOR.

The particle sizes of the RO200R-samples were investigated using STEM-EDX (Fig. 1g–i). The spatial distribution of the particles over the supports was not markedly different from that in the R-samples (Fig. 1a–c). Furthermore, the average size of the cobalt particles was similar to that of the R-samples, despite significantly different values obtained by H$_2$-chemisorption (Fig. 3b–d, Table 1). In fact, the theoretical H$_2$-uptake now approached the values of the experimental H$_2$-uptake.

On TiO$_2$, the increase in H$_2$-uptake with a similar average particle size implied that ROR did not induce substantial redispersion of cobalt. Rather, the ROR treatment resulted in a higher fraction of accessible metallic cobalt surface through a decrease of the surface coverage by suboxide species (Fig. 3e). On Nb$_2$O$_5$, however, the situation is more complex, because the experimental H$_2$-uptake after ROR was higher than the

theoretical one. The additional H$_2$-uptake could be rationalized in two ways. First, redispersion of few large cobalt particles could have formed small particles, which would be undetectable by STEM-EDX. Second, hydrogen could have spilled over to Nb$_2$O$_5$, in line with the fact that Nb$_2$O$_5$ has been proven to facilitate hydrogen storage[34,35]. On α-Al$_2$O$_3$, the same Kirkendall behavior occurred at oxidation temperatures above 100 °C (Fig. 1f), but here the H$_2$-uptake decreased on all ROR-samples due to particle growth.

ROR has been reported to regenerate deactivated catalysts through redispersion of the active metal via the Kirkendall effect[26–29]. However, the work on catalyst regeneration focused on cobalt catalysts supported on irreducible oxides, such as Al$_2$O$_3$ and SiO$_2$, in which SMSI effects do not play a role. Furthermore, we ascribe the absence of net redispersion in our samples to different initial cobalt particle sizes. The particles in our R-TiO$_2$ (11 nm) and R-Al$_2$O$_3$ (13 nm) were substantially smaller than those reported in literature (22 nm)[29]. According to Sadasivan et al.[36], only cobalt particles above a certain critical size give rise to redispersion upon ROR treatments.

The trends observed with $H_2$-chemisorption for the reducible oxides can be understood based on the previously discussed results. The Kirkendall effect was most likely key in tuning the SMSI during ROR, because at oxidation temperatures up to 100 °C, where this effect was not observed, the increase in $H_2$-uptake was minor. The $H_2$-uptake only increased significantly when the Kirkendall effect took place. On the other side of the optimum, at an oxidation temperature of 400 °C, the high temperature induced particle growth from 10 up to 15 nm for RO400R-$TiO_2$, as determined using STEM-EDX (Supplementary Figure 5). This reduced the metallic surface area and counteracted the positive effect of the ROR treatment on the SMSI.

Further indications for the optimization of SMSI effects through ROR were found when the second reduction was performed at a higher temperature (Fig. 4). The $H_2$-uptake decreased for the RO200R-$TiO_2$ and RO200R-$Nb_2O_5$ samples with increasing reduction temperature. At 350 °C, the $H_2$-uptake values returned to the values of the initial R-samples that were reduced once. Apparently, the mild temperature of the second reduction was essential to keep the concentration or mobility of the suboxide species low, while it was sufficiently high to reduce cobalt oxide to metallic cobalt. Additionally, the $H_2$-uptakes of the R-samples at reduction temperatures of 220 and 350 °C were comparable. The substantial increase in $H_2$-uptake for the RO200R-samples was thus not merely caused by reduction at lower temperature. In contrast, the $H_2$-uptake of the Co/α-$Al_2O_3$ catalysts increased slightly at higher reduction temperatures.

**Fischer-Tropsch**. The catalytic activity of Co/$TiO_2$ and Co/$Nb_2O_5$ in FT after ROR followed the same trend as $H_2$-chemisorption, i.e., increased activity with the optimum at an oxidation temperature of 200 °C (Fig. 5a). Notably, the activity doubled on $TiO_2$ and $Nb_2O_5$ as a result of the RO200R treatment and the enhancement was stable over at least 100 h time-on-stream (Supplementary Figure 6). As expected, the activity of Co/α-$Al_2O_3$ was not enhanced, indicating that the positive effect of the ROR treatment was specific for catalysts supported on reducible oxides. The $C_{5+}$-selectivity for all catalysts was 85–89% and did not change significantly as a result of the ROR treatments (Supplementary Table 3).

The observed cobalt-weight-based activities of RO200R-$TiO_2$ ($16 \times 10^{-5}$ $mol_{CO}$ $g_{Co}^{-1}$ $s^{-1}$) and RO200R-$Nb_2O_5$ ($8.9 \times 10^{-5}$ $mol_{CO}$ $g_{Co}^{-1}$ $s^{-1}$) are exceptionally high. Compared to literature[12,30,37–41], activities in this range are normally only achieved by promoting a cobalt-based catalyst with noble metals (Supplementary Figure 7). Tuning of the metal–support interactions by ROR or other pretreatments might thus provide an attractive alternative to promotion with expensive and scarce noble metals.

The ROR treatments increased the catalytic activity of Co/$TiO_2$ and Co/$Nb_2O_5$ with respect to the R-samples. The increase in activity was proportional to the $H_2$-uptake (Fig. 5b), indicating that the surface-specific activity or turnover frequency (TOF, based on activity and $H_2$-chemisorption data; see SI for details) was constant and more surface sites with identical characteristics became available. The opposite was observed with irreducible α-$Al_2O_3$, where the activity decreased after ROR and less surface sites were available albeit at constant TOF. Clearly, the TOF was support-dependent and increased in the order α-$Al_2O_3$ < $Nb_2O_5$ < $TiO_2$.

The high TOF values for the reducible oxides show that Co surface sites were promoted by $TiO_x$ or $NbO_x$ in the R- and ROR-samples. In addition, the TOF was independent of ROR, meaning that ROR did not change the nature of the promoted active sites. Therefore, the enhanced cobalt-weight-based activity of Co/$TiO_2$

and Co/$Nb_2O_5$ implied that the number of promoted active sites increased as a result of ROR. Furthermore, the effect of the second reduction temperature on the activity is included as well in Fig. 5b for Co/$Nb_2O_5$. Reduction at 250 °C gave only a moderate increase in $H_2$-chemisorption and a corresponding increase in activity, while reduction at 220 °C doubled the $H_2$-uptake and activity.

Although the TOF of cobalt catalysts has been reported to be support independent[37,42], recent publications have shown that the promoting effect of the support can be substantial[17,18,21,30]. This promoting effect on the TOF has been related to the Lewis acidity of the metal oxide: an increase in acidity leads to an increase in TOF[17,18]. The Lewis acid character of oxidic materials originates from exposed metal centers[43], meaning that the structural arrangement and oxidation state of the ions will determine their acid strength[44]. Jeong et al.[45] have recently reported an acidity scale based on the intramolecular charge transfer energy between organic molecules and several metal oxides (Supplementary Table 4). The values reported for the supports used in our work do predict higher TOF for $TiO_2$ and $Nb_2O_5$ compared to $Al_2O_3$, but the comparison between $TiO_2$ and $Nb_2O_5$ requires further analysis. As mentioned before, Lewis acidity depends on the oxidation state of the cation, thus values for lower oxidation states in suboxides (e.g., $Ti^{3+}$, $Nb^{4+}$, or $Nb^{3+}$) gain importance for reducible metal oxides as shown in the work of Boffa et al.[21,46]. Because the oxidation state and structural arrangement of the relevant suboxides are not completely established, the formal Lewis acidity remains uncertain.

In summary, ROR treatments were investigated as a tool to tune metal-support interactions in supported cobalt catalysts to improve their performance in the FT synthesis. Reduction of cobalt oxide supported on reducible ($TiO_2$ and $Nb_2O_5$) and irreducible (α-$Al_2O_3$) oxides followed by controlled oxidation led to hollow cobalt oxide particles via the Kirkendall effect. A second reduction resulted in a twofold increase in $H_2$-uptake for the $TiO_2$- and $Nb_2O_5$-supported samples and a decrease for the α-$Al_2O_3$-supported sample. This effect was attributed to modulation of the metal–support interactions rather than redispersion of the cobalt phase. The modified metal–support interactions greatly increased the catalytic activity in the FT synthesis, as the activity doubled on $TiO_2$ and $Nb_2O_5$ and decreased slightly on α-$Al_2O_3$. This increased activity was caused by more accessible cobalt surface sites as revealed by the constant turnover frequency, while preserving the full promotional effect of the $TiO_x$ or $NbO_x$ species. The more efficient utilization of the cobalt and support interphase led to exceptionally active catalysts. Tailoring of metal-support interactions via ROR treatments is thus an effective method to improve the activity of metal catalysts supported on reducible oxides.

## Methods

**Synthesis**. The samples were synthesized by incipient wetness impregnation using aqueous cobalt nitrate solutions. The precursor solutions were prepared by dissolving appropriate amounts of cobalt nitrate hexahydrate (99+%, Acros Organics) in Milli-Q water to obtain 6 wt% cobalt loading on each catalyst. The supports, $TiO_2$ (P25, Evonik, $SA_{BET}$: 50 $m^2$ $g^{-1}$, PV: 0.30 $cm^3$ $g^{-1}$), $Nb_2O_5$ (pseudo-hexagonal TT-phase, CBMM, $SA_{BET}$: 9 $m^2$ $g^{-1}$, PV: 0.06 $cm^3$ $g^{-1}$), and α-$Al_2O_3$ (BASF, $SA_{BET}$: 7 $m^2$ $g^{-1}$, PV: 0.02 $cm^3$ $g^{-1}$), were sieved to a grain size of 75–150 μm. $Nb_2O_5$ was obtained from amorphous niobium oxide hydrate (HY-340, AD/4465, CBMM) by calcination in a muffle oven at 600 °C for 4 h.

For a typical impregnation, 2 g of support was pre-dried at 80 °C for 1 h under vacuum. Just before the impregnation, the vacuum was released and a pore filling amount of precursor solution was added dropwise to the support under magnetic and manual stirring. In the case of Co/$TiO_2$, 2 g of sample was dried and heat treated in fluidized bed mode in a U-shaped glass reactor following a previously reported procedure[28]. The sample was first dried at 80 °C for 2 h and heat treated directly afterwards at 250 °C for 2 h. Both steps were performed in 1 L $min^{-1}$ $N_2$ upward flow with a heating ramp of 2 °C $min^{-1}$. Co/$Nb_2O_5$ and Co/α-$Al_2O_3$ catalysts were dried for 12 h at 60 °C overnight in stagnant air and subsequently

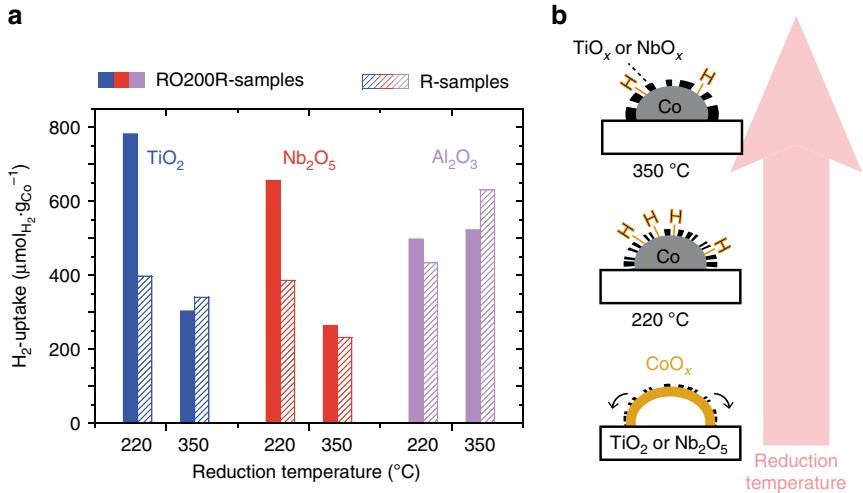

**Fig. 4** The influence of the temperature of the second reduction on the exposed cobalt surface area. **a** Hydrogen uptake as a function of the second reduction temperature. Dashed bars correspond to the pristine samples after reduction at 220 or 350 °C and solid filled bars correspond to the samples after reduction at 350 °C followed by oxidation at 200 °C and reduction at 220 or 350 °C. **b** Schematic illustration of the effect of the reduction temperature on catalysts supported on reducible oxides

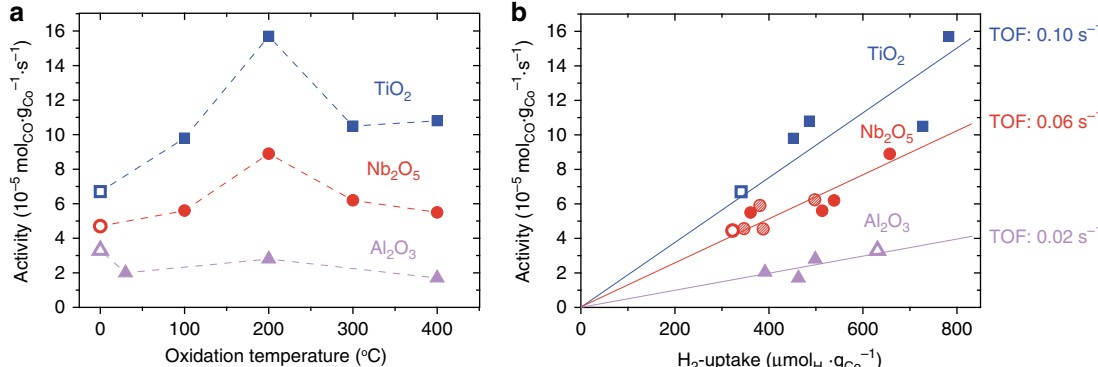

**Fig. 5** The effect of the ROR treatment on the catalytic activity in the Fischer–Tropsch synthesis. FT was performed at 20 bar, 220 °C, $H_2/CO = 2$ $V/V$, GHSV = 1000–13000 $h^{-1}$, time-on-stream = 90–100 h, CO conv. = 15–34% ($TiO_2$), 17–25% ($Nb_2O_5$), and 8–12% ($Al_2O_3$). **a** Cobalt-weight-based catalytic activity of $Co/TiO_2$ (blue squares), $Co/Nb_2O_5$ (red circles), and $Co/Al_2O_3$ (violet triangles) after various reduction–oxidation–reduction treatments. The activity of the pristine samples after reduction at 350 °C (R-samples, open symbols) is included at 0 on the x-axis. **b** Cobalt-weight-based activity as a function of hydrogen uptake for the R-samples (open symbols) and the samples after reduction at 350 °C followed by oxidation between 30 and 400 °C and reduction at either 220 °C (filled symbols) or at 250 °C (dashed symbols); values for the turnover frequencies (TOF) have been calculated from the slopes of the lines

calcined in a fixed bed reactor under $N_2$ flow (1 L min$^{-1}$) at 350 °C for 2 h with a heating ramp of 3 °C min$^{-1}$. Multiple batches of 2 g were mixed to obtain a substantial amount of sample. The obtained samples were pressed and sieved to a grain size of 75–150 μm.

Ex situ RO treatments were performed in U-shaped glass reactors. To this end, batches of 500 mg were first reduced at 350 °C for 2 h (2 °C min$^{-1}$) in 120 mL min$^{-1}$ of 25 vol% $H_2$ in $N_2$ and subsequently cooled to 100 °C. At this point, the reactor was flushed with $N_2$ and cooled down further to 30 °C. The oxidation step followed directly after the reduction and was performed in 95 mL min$^{-1}$ of 5 vol% $O_2$ in $N_2$ at various temperatures up to 400 °C for 2 h with a 2 °C min$^{-1}$ heating ramp.

**Characterization**. STEM combined with high-angle annular dark-field (STEM-HAADF) and EDX analyses were performed on an FEI Talos F200X. This microscope was equipped with a high-brightness field emission gun (X-FEG) and a Super-X G2 EDX detector and operated at 200 kV. The EM samples were prepared by suspending the samples in 2-propanol (>99.9%, Sigma-Aldrich) using sonication and dropcasting the suspension on a carbon-coated Cu grid (200 mesh). The cobalt particle size was determined using the iTEM software and at least 200 particles were analyzed. The reduced and passivated samples were corrected for a 3 nm CoO layer and the particle surface averaged diameters (D[3,2]) were calculated.

$H_2$-chemisorption was measured on a Micromeritics ASAP 2020 C using ~200 mg of sample. The catalysts were reduced in pure $H_2$ prior to the measurement. The pristine samples were reduced for 2 h (1 °C min$^{-1}$) at 350 °C, while the RO-samples were reduced at 220 °C. After reduction, the samples were evacuated, cooled to 150 °C and analyzed at that temperature. The metallic surface area and average particle diameter were calculated assuming a cobalt cross-sectional area of 0.0662 nm$^2$, an H/Co stoichiometry of 1 and spherical cobalt particles[26].

Inductively coupled plasma-optical emission spectroscopy was performed on a SPECTRO ARCOS. The samples were prepared by extraction of cobalt using aqua regia.

TPR profiles were measured using a Micromeritics Autochem 2920, equipped with a TCD detector. Typically, 100 mg sample was reduced up to 1000 °C with 5 °C min$^{-1}$ in a flow of 5 vol% $H_2$ in Ar. To determine the degree of reduction, the samples were first reduced in the TPR setup in a flow of pure $H_2$ for 2 h at 350 or 220 °C depending of the stage in the ROR treatment and then, the TPR profile was recorded from 50 to 800 °C in a flow of 5 vol% $H_2$ in Ar. The cobalt that had not been reduced was quantified from the areas of the peaks between 250 and 700 °C.

**Fischer-Tropsch**. The catalytic performance was measured using an Avantium Flowrence 16 parallel reactor setup. Stainless steel reactors (ID = 2 mm) were loaded with 75–100 mg of catalyst (75–150 μm), and diluted with 100 mg SiC (212–425 μm). Prior to reaction, the catalysts were reduced in situ for 8 h

(1 °C min$^{-1}$) in 25 vol% $H_2$ in He at 1 bar. The pristine samples were reduced at 350 °C, while the RO-samples were reduced at 220 °C. After reduction, the temperature was lowered to 180 °C (3 °C min$^{-1}$) and the pressure was increased to 20 bar under $H_2$ flow. Finally, synthesis gas with $H_2$/CO = 2 $V/V$ (5 vol% He as internal standard) was introduced and the temperature was raised to 220 °C with 1 °C min$^{-1}$. The product stream was analyzed online with an Agilent 7890A GC. The hydrocarbon products were analyzed on an Agilent J&W PoraBOND Q column connected to an FID. The permanent gasses were separated on a ShinCarbon ST column and quantified using a TCD. Definitions of the selectivity and activity, expressed as CO conversion, cobalt time yield and TOF, are included as Supplementary Methods.

## Data availability

The datasets generated and/or analyzed during the current study are available from the corresponding author on reasonable request.

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

## Acknowledgments

Companhia Brasileira de Metalurgia e Mineração (CBMM), Shell Global Solutions, and the Netherlands Association for Scientific Research (NWO) are thanked for financial support. K.P.d.J. acknowledges the European Research Council, EU FP7 ERC Advanced Grant no. 338846. Robson Monteiro (CBMM), Rogério Ribas (CBMM), and Peter Munnik (Shell Global Solutions) are acknowledged for useful discussions. Wouter Lamme, Jessi van der Hoeven, Nynke Krans and Mark Meijerink (STEM-EDX), and Helen de Waard (ICP-OES) are acknowledged for the measurements indicated.

## Author contributions

C.H.M. and T.W.v.D. contributed equally to the synthesis, characterization, evaluation of the catalytic performance and writing of the manuscript; K.P.d.J. contributed to experimental design, data analysis, and manuscript writing.

## Additional information

**Competing interests:** The authors declare no competing interests.

