## [Peer Review File · Nature Communications]

Reviewers' comments:

Reviewer #1 (Remarks to the Author):

Increasing the activity of Fischer-Tropsch synthesis (FTS) catalysts has long been considered as a major challenge for both cobalt based and iron based catalysts. For cobalt based catalysts, some methods, involving crystalline phase transformation (from fcc to hcp phase), modification of the CO adsorption and dissociation by the addition of metal oxide promoters and using noble metals to enhance the hydrogen spillover process have been proved efficient. The ROR process reported in this manuscript has been proposed earlier by Kobylinski et al. in the 1980s (e.g. patents US 4605679 and US 4605676), and to my knowledge, other researchers have used it to tune the activity and selectivity of varying cobalt based catalysts. Although metal oxides such as Nb₂O₅ and TiO₂ may not have been investigated, I am afraid that this manuscript may not possess enough novelty to meet the requirement of Nature Communications.

On the other hand, more evidences, especially theoretical calculations (e.g. DFT calculations) should be provided by the authors to illustrate the interaction mechanism between the metal and supports before and after ROR treatments. Besides, perhaps some operando methods, such as in situ XPS should be rather involved to reveal the change of surface elemental composition during the ROR process, and TEM graphs should also be provided, at least in the supporting information, to show the microscopy of Co nanoparticles more clearly. Moreover, what is the crystalline phase of Co before and after ROR process? I suggest that XRD patterns should be provided as well. Generally, in my opinion, this manuscript is better to be published on a journal such as ACS catalysis and Journal of Catalyst as a research article. Considering its novelty and thoroughly investigations, I look forward seeing its fast publication and being cited by other researchers.

Reviewer #2 (Remarks to the Author):

This manuscript reports an interesting phenomenon that the treatment of Co catalysts loaded on reducible metal oxide support by reduction-oxidation-reduction (ROR) can significantly enhance the H₂ chemisorption and catalytic activity for Fischer-Tropsch synthesis. It has been clarified that the ROR changes the interaction between Co and reducible metal oxides, thus enhancing the accessible Co sites and the catalytic activity per gram of Co. The activity per Co site remains almost unchanged. In my opinion, this is an interesting contribution to understanding the activity change during different catalyst treatments and may guide the catalyst design for FT synthesis and the related reactions using similar types of catalysts. The manuscript is also well organized. Therefore, I recommend the acceptance of this manuscript without change.

Reviewer #3 (Remarks to the Author):

This study nicely reports a rational strategy for significantly enhancing the activity for Fischer-Tropsch synthesis (FTS) of cobalt catalysts (ca. 6 wt% loading) supported on reducible oxides, exhibiting the known SMSI state and exemplified here for TiO₂ and Nb₂O₅, based on submitting the impregnated and calcined samples to reduction-oxidation-reduction (ROR) treatments prior catalysis. Equivalent Co catalysts supported on non-reducible alpha-Al₂O₃ were also prepared for comparison purposes. ROR treatments at suitable conditions lead to about a two-fold increase in activity per total mass of cobalt (cobalt-time-yield) without substantially modifying the metal dispersion (STEM-EDX) nor the intrinsic activity (TOF, based on H₂ chemisorption) of the surface metal cobalt sites.

Given its scientific and practical relevance to the design of more efficient Co-based FTS catalysts, the work merits publication in Nature Commun. provided the following minor issues are properly addressed:

1. The degrees of cobalt reduction (DOR) for the R- and ROR-samples have to be given and taken into account for calculating TOFs. For example, it may be expected from the H₂-TPR profiles shown in Fig. 2a for Co/Nb₂O₅ that not all cobalt oxides were reduced to the metallic state after submitting the calcined (Co₃O₄/Nb₂O₅) and the RO200 and RO400 samples to reduction at 350 °C and 220 °C, respectively.

2. The authors state in page 9 that “the high TOF values for the reducible oxides show that Co surface sites were promoted by TiO_x and NbO_x in the optimized SMSI state after ROR”. However, TOFs were concluded to be unaffected by ROR treatments and, therefore, the higher TOF of TiO₂- and Nb₂O₅-supported catalysts compared to those supported on alpha-Al₂O₃ is intrinsic to the chemical identity of the support irrespective of whether or not the SMSI state in the former catalysts is optimized through ROR treatments.

3. Re-oxidation of decorating TiO_x and NbO_x species by water formed as by-product during FTS may occur, partly reverting the SMSI state under reaction conditions and, consequently, increasing the amount of surface metal sites available for reaction. Did such partial reversion of the SMSI state occur for Co/TiO₂ and Co/Nb₂O₅ under FTS conditions? How could this fact influence the pseudo-steady state TOFs calculated from H₂ chemisorption measured on pre-reacted samples and, thus, the conclusions of this work regarding the invariability of TOFs for a given support?

4. Additional minor issues:

- Abstract, line 4 from bottom: Fischer (instead of Fisher).

- How accurate were the carbon mass balances in the reported catalytic experiments? This is a critical issue in FTS as poor carbon mass balances may strongly affect the reported hydrocarbon selectivity.

Response to reviewers

Manuscript NCOMMS-18-17963-T

Black	Reviewer's comments
Blue, italics	Author's response
Blue, italics, underlined	Changes to the manuscript/supplementary information

All references to page numbers or figure/table captions in this rebuttal letter apply to the revised version of the manuscript and supplementary information.

Reviewers' comments:

Reviewer #1 (Remarks to the Author):

Increasing the activity of Fischer-Tropsch synthesis (FTS) catalysts has long been considered as a major challenge for both cobalt based and iron based catalysts. For cobalt based catalysts, some methods, involving crystalline phase transformation (from fcc to hcp phase), modification of the CO adsorption and dissociation by the addition of metal oxide promoters and using noble metals to enhance the hydrogen spillover process have been proved efficient. The ROR process reported in this manuscript has been proposed earlier by Kobylinski et al. in the 1980s (e.g. patterns US 4605679 and US 4605676), and to my knowledge, other researchers have used it to tune the activity and selectivity of varying cobalt based catalysts. Although metal oxides such as Nb₂O₅ and TiO₂ may not have been investigated, I am afraid that this manuscript may not possess enough novelty to meet the requirement of Nature Communications.

We thank the reviewer for the analysis of our work and the provided feedback.

We agree with the reviewer that the ROR process itself is not new and has been reported before in the 1980's patent literature as well more recent work by Sasol, which we referred to in the introduction. In these cases, ROR was applied to cobalt catalysts on irreducible oxides to re-disperse the cobalt, thereby forming smaller particles and increasing catalytic activity. However, we kindly disagree with the reviewer about the novelty of the present manuscript, as the novelty is in the application of ROR to cobalt on reducible oxides and the observations that ROR tuned the metal-support interactions, rather than re-dispersed the cobalt phase, and its pronounced enhancement of the catalytic performance. Therefore, we feel that this manuscript presents sufficient novelty to warrant publication in Nature Communications.

On the other hand, more evidences, especially theoretical calculations (e.g. DFT calculations) should be provided by the authors to illustrate the interaction mechanism between the metal and supports before and after ROR treatments.

We are not aware of theoretical methods to model the metal-support interactions that occur with reducible metal oxides during reduction and/or oxidation of the catalysts. The phenomena that we discovered occur at the mesoscale which would leave DFT calculations most likely without success.

Besides, perhaps some operando methods, such as in situ XPS should be rather involved to reveal the change of surface elemental composition during the ROR process,

So-called Near-Ambient Pressure XPS (NAP-XPS) has been performed at the synchrotron facility BESSY II to verify whether XPS can be applied to investigate the metal-support interactions. A pre-reduced and passivated Co/Nb₂O₅ sample was heated to 400 °C in 0.5 mbar H₂. Below we provide an example of the Co2p signal measured using a kinetic energy of 800 eV at 400 °C in H₂. The peak at binding energy (BE) 778 eV was attributed to Co⁰, but the shoulders and satellite peaks at higher BE clearly indicate that a substantial amount of Co²⁺ was still present, even at higher temperature. The operating conditions were thus not sufficient to reduce cobalt completely, which implies that any result would not be fully representative of the actual sample during ROR.

Furthermore, only Nb⁵⁺ was observed both before and after reduction during a depth profile measurement with different kinetic energies, meaning that suboxide formation could not be observed. Based on these experiences, NAP-XPS was found unsuitable to address our research question and therefore we decided not to include it in the manuscript.

and TEM graphs should also be provided, at least in the supporting information, to show the microscopy of Co nanoparticles more clearly.

Additional microscopy results were added to the supplementary information and referenced to in the main text.

We included in the main text on p4: “Additional electron microscopy results for these samples can be found in Supplementary Figures 1-3.”

We added to the supplementary information on p4-6:

“Supplementary Figure 1. Additional electron microscopy images of the TiO₂-supported samples upon reduction-oxidation-reduction treatments. STEM-HAADF combined with EDX mapping of Co/TiO₂ at various stages of the ROR treatment. Co is depicted in yellow. a, b, c, d, Samples after reduction at 350 °C and passivation (R-TiO₂); e, f, g, h, samples after reduction at 350 °C and oxidation at 200 °C (RO200-TiO₂); i, j, k, l, samples after reduction at 350 °C, oxidation at 200 °C and reduction at 220 °C (RO200R-TiO₂) and passivation.

Supplementary Figure 2. Additional electron microscopy images of the Nb₂O₅-supported samples upon reduction-oxidation-reduction treatments. STEM-HAADF combined with EDX mapping of Co/Nb₂O₅ at various stages of the ROR treatment. Co is depicted in yellow. a, b, c, d, Samples after reduction at 350 °C and passivation (R- Nb₂O₅); e, f, g, h, samples after reduction at 350 °C and oxidation at 200 °C (RO200- Nb₂O₅); i, j, k, l, samples after reduction at 350 °C, oxidation at 200 °C and reduction at 220 °C (RO200R- Nb₂O₅) and passivation.

Supplementary Figure 3. Additional electron microscopy images of the Al₂O₃-supported samples upon reduction-oxidation-reduction treatments. a, d, g, h, i, STEM-HAADF and b, c, e, f, bright-field TEM of Co/Al₂O₃ at various stages of the ROR treatment. a, b, c, Samples after reduction at 350 °C and passivation (R- Al₂O₃); d, e, f, samples after reduction at 350 °C and oxidation at 200 °C (RO200- Al₂O₃); g, h, i, samples after reduction at 350 °C, oxidation at 200 °C and reduction at 220 °C (RO200R- Al₂O₃) and passivation.

Moreover, what is the crystalline phase of Co before and after ROR process? I suggest that XRD patterns should be provided as well.

*We performed multiple XRD experiments on Co/TiO₂ and Co/Nb₂O₅ both before and after ROR treatments, including in situ XRD under H₂ up to 350 °C to mimic a reduction treatment. As an example, a diffraction pattern of Co/TiO₂ after reduction at 350 °C and passivation (RO30) is provided below. However, cobalt (oxide) was not observed in all these measurements because of the low cobalt loading applied here (6 wt%). This is in line with earlier work in our group on similar 6 wt% Co/Nb₂O₅ where the absence of a cobalt (oxide) signal before and during reduction was attributed to poor crystallinity in these catalysts (den Otter et al., J. Catal. **340**, 270-275 (2016)). In addition, detection of the cobalt (oxide) signal on Nb₂O₅ is further complicated by overlap with the diffraction pattern of Nb₂O₅.*

Because XRD gave no information about the crystal structure of Co, we decided not to include it in the manuscript or supplementary information.

Generally, in my opinion, this manuscript is better to be published on a journal such as ACS catalysis and Journal of Catalyst as a research article. Considering its novelty and thoroughly investigations, I look forward seeing its fast publication and being cited by other researchers.

Based on the potential benefit of the present findings to various applications other than catalysis and the considerations described before concerning the novelty, we are of the opinion that this manuscript is suitable for publication in a general natural sciences journal, such as Nature Communications also following the favorable recommendations of reviewers #2 and 3.

Reviewer #2 (Remarks to the Author):

This manuscript reports an interesting phenomenon that the treatment of Co catalysts loaded on reducible metal oxide support by reduction-oxidation-reduction (ROR) can significantly enhance the H₂ chemisorption and catalytic activity for Fischer-Tropsch synthesis. It has been clarified that the ROR changes the interaction between Co and reducible metal oxides, thus enhancing the accessible Co sites and the catalytic activity per gram of Co. The activity per Co site remains almost unchanged. In my opinion, this is an interesting contribution to understanding the activity change during different catalyst treatments and may guide the catalyst design for FT synthesis and the related reactions using similar types of catalysts. The manuscript is also well organized. Therefore, I recommend the acceptance of this manuscript without change.

We thank the reviewer for examining our manuscript and his/her positive recommendation.

Reviewer #3 (Remarks to the Author):

This study nicely reports a rational strategy for significantly enhancing the activity for Fischer-Tropsch synthesis (FTS) of cobalt catalysts (ca. 6 wt% loading) supported on reducible oxides, exhibiting the known SMSI state and exemplified here for TiO₂ and Nb₂O₅, based on submitting the impregnated and calcined samples to reduction-oxidation-reduction (ROR) treatments prior catalysis. Equivalent Co catalysts supported on non-reducible alpha-Al₂O₃ were also prepared for comparison purposes. ROR treatments at suitable conditions lead to about a two-fold increase in activity per total mass of cobalt (cobalt-time-yield) without substantially modifying the metal dispersion (STEM-EDX) nor the intrinsic activity (TOF, based on H₂ chemisorption) of the surface metal cobalt sites.

Given its scientific and practical relevance to the design of more efficient Co-based FTS catalysts, the work merits publication in Nature Commun. provided the following minor issues are properly addressed:

We thank the reviewer for his/her attention to our work and the provided feedback.

1. The degrees of cobalt reduction (DOR) for the R- and ROR-samples have to be given and taken into account for calculating TOFs. For example, it may be expected from the H₂-TPR profiles shown in Fig. 2a for Co/Nb₂O₅ that not all cobalt oxides were reduced to the metallic state after submitting the calcined (Co₃O₄/Nb₂O₅) and the RO200 and RO400 samples to reduction at 350 °C and 220 °C, respectively.

The DOR has been determined by TPR of the R- and the RO200R-samples, and of RO400R-Nb₂O₅. For all these samples, the DOR was between 84-98 %, indicating that cobalt oxide was almost completely reduced, regardless of the ROR treatment. Because the DOR was close to 100 % and the differences between the samples were small, we decided not to correct the H₂-uptake (and hence TOF) values.

We added to the main text on p6: “The degree of reduction (DOR) was determined by TPR directly after the last reduction for all R- and RO200R-samples and for RO400R-Nb₂O₅ (Supplementary Fig. 4 and Supplementary Table 2). The DOR for the R-samples was 97-98 % and decreased slightly to 84-95 % after ROR. Because these changes were minor, the H₂-chemisorption results were not corrected for the DOR.”

We added to the experimental methods in the main text on p11: “To determine the degree of reduction, the samples were first reduced in the TPR setup in a flow of pure H₂ for 2 h at 350 °C or 220 °C depending of the stage in the ROR treatment and then, the TPR profile was recorded from 50 to 800 °C in a flow of 5 vol.% H₂ in Ar. The cobalt that had not been reduced was quantified from the areas of the peaks between 250 and 700 °C.”

We added to the supplementary information on p7:

Supplementary Figure 4. TPR profiles used to determine the degree of reduction of cobalt. The baseline is shown for the peaks that were integrated as cobalt that had not been reduced and thus lowered the degree of reduction. The degree of reduction was determined after reduction at 350 °C (R-samples), after reduction at 350 °C, oxidation at 200 °C and reduction at 220 °C (RO200R-samples) and after reduction at 350 °C, oxidation at 400 °C and reduction at 220 °C (RO400R-Nb₂O₅).

Supplementary Table 2. The degree of cobalt reduction at various stages of the ROR treatment, as measured by TPR. The degree of reduction was determined after reduction at 350 °C (R-samples), after reduction at 350 °C, oxidation at 200 °C and reduction at 220 °C (RO200R-samples) and after reduction at 350 °C, oxidation at 400 °C and reduction at 220 °C (RO400R-Nb₂O₅).

Sample	DOR (%)		
	R	RO200R	RO400R
Co/TiO ₂	97	86	n.d.
Co/Nb ₂ O ₅	98	93	84
Co/Al ₂ O ₃	98	95	n.d.

2. The authors state in page 9 that “the high TOF values for the reducible oxides show that Co surface sites were promoted by TiO_x and NbO_x in the optimized SMSI state after ROR”. However, TOFs were concluded to be unaffected by ROR treatments and, therefore, the higher TOF of TiO₂- and Nb₂O₅-supported catalysts compared to those supported on alpha-Al₂O₃ is intrinsic to the chemical identity of the support irrespective of whether or not the SMSI state in the former catalysts is optimized through ROR treatments.

Indeed, the TOF is independent of ROR treatments for all catalysts. We agree with the reviewer that the variations in TOF values for the different supports are caused by the chemical nature of the support, which also includes promotion by suboxides as part of the chemistry of certain metal nanoparticles on reducible oxides under reducing conditions.

In order to strengthen this point and to avoid confusion, we changed the concerning paragraph on p9 to: “The high TOF values for the reducible oxides show that Co surface sites were promoted by TiO_x or NbO_x in the R- and ROR-samples. In addition, the TOF was independent of ROR, meaning that ROR did not change the nature of the promoted active sites. Therefore, the enhanced cobalt-weight-based activity of Co/TiO₂ and Co/Nb₂O₅ implied”

that the number of promoted active sites increased as a result of ROR. Furthermore, the effect of the second reduction temperature on the activity is included as well in Fig. 5b for Co/Nb₂O₅. Reduction at 250 °C gave only a moderate increase in H₂-chemisorption and a corresponding increase in activity, while reduction at 220 °C doubled the H₂-uptake and activity.”

3. Re-oxidation of decorating TiO_x and NbO_x species by water formed as by-product during FTS may occur, partly reverting the SMSI state under reaction conditions and, consequently, increasing the amount of surface metal sites available for reaction. Did such partial reversion of the SMSI state occur for Co/TiO₂ and Co/Nb₂O₅ under FTS conditions? How could this fact influence the pseudo-steady state TOFs calculated from H₂ chemisorption measured on pre-reacted samples and, thus, the conclusions of this work regarding the invariability of TOFs for a given support?

We agree that a positive effect of ROR treatments via tuning the SMSI state on the catalytic performance might a priori be unexpected given the FT reaction conditions, e.g. high steam partial pressures. However, we obtained a linear correlation between H₂-uptake from H₂-chemisorption and catalytic activity (Figure 5b), showing that the enhanced metallic Co surface area after RO200R was preserved during FT with a constant TOF. Furthermore, stable activity was observed for 100 h on-stream (see Supplementary Figure 2), which indicated that the modified SMSI state remained unchanged on the time scale of our catalytic experiments. These two observations show that the SMSI state was not reversed during FT so much as to annihilate the positive effect of the ROR treatments on the catalytic activity.

This point was already addressed in the manuscript on p8: “Notably, the activity doubled on TiO₂ and Nb₂O₅ as a result of the RO200R treatment and the enhancement was stable over at least 100 h time-on-stream (Supplementary Fig. 2).” In addition, this point is further clarified by the modifications discussed under point 2.

4. Additional minor issues:

- Abstract, line 4 from bottom: Fischer (instead of Fisher).

We changed in the main text on p1 “Fisher” to “Fischer”

- How accurate were the carbon mass balances in the reported catalytic experiments? This is a critical issue in FTS as poor carbon mass balances may strongly affect the reported hydrocarbon selectivity.

The carbon mass balance was not determined in full in our catalytic experiments, due to the design of the setup (high-throughput) and its analytics, using an internal standard and quantifying only hydrocarbon products up to C₉. However, given that no byproducts, such as CO₂ or short-chain alcohols, were observed under these reaction conditions, the carbon balance is expected to be close to 100 %.

REVIEWERS' COMMENTS:

Reviewer #3 (Remarks to the Author):

In the revised version the authors have properly addressed the issues raised by this reviewer. In my opinion, the manuscript is now ready to be accepted for publication in Nature Commun. without further changes.